# Recurrent loss of CenH3 is associated with independent transitions to holocentricity in insects

**Ines A Drinnenberg[1], Dakota deYoung[2], Steven Henikoff[1]\*, Harmit Singh Malik[1,3]\***

[1]Division of Basic Sciences, Fred Hutchinson Cancer Research Center, Seattle, United States; [2]Department of Biology, University of Washington, Seattle, United States; [3]Howard Hughes Medical Institute, Fred Hutchinson Cancer Research Center, Seattle, United States

**Abstract** Faithful chromosome segregation in all eukaryotes relies on centromeres, the chromosomal sites that recruit kinetochore proteins and mediate spindle attachment during cell division. The centromeric histone H3 variant, CenH3, is the defining chromatin component of centromeres in most eukaryotes, including animals, fungi, plants, and protists. In this study, using detailed genomic and transcriptome analyses, we show that CenH3 was lost independently in at least four lineages of insects. Each of these lineages represents an independent transition from monocentricity (centromeric determinants localized to a single chromosomal region) to holocentricity (centromeric determinants extended over the entire chromosomal length) as ancient as 300 million years ago. Holocentric insects therefore contain a CenH3-independent centromere, different from almost all the other eukaryotes. We propose that ancient transitions to holocentricity in insects obviated the need to maintain CenH3, which is otherwise essential in most eukaryotes, including other holocentrics.

**\*For correspondence:** steveh@fhcrc.org (SH); hsmalik@fhcrc.org (HSM)

**Competing interests:** The authors declare that no competing interests exist.

## Introduction

In eukaryotes, accurate chromosome segregation relies on specific chromosomal regions called centromeres that recruit components of the proteinaceous kinetochore complex to mediate spindle attachments and ensure high-fidelity segregation (*DeWulf and Earnshaw, 2008*). In animals, fungi, and plants, kinetochore assembly onto centromeres occurs in a hierarchical process that strictly depends on the presence of the specialized histone H3 variant CenH3 (first identified as Cenp-A in mammals [*Earnshaw and Rothfield, 1985*; *Palmer et al., 1991*]), which replaces the canonical H3 in centromeric nucleosomes (*Sullivan et al., 1994*; *Yoda et al., 2000*). In all organisms that have been studied, CenH3 deletions are lethal and lead to catastrophic defects in chromosome segregation (*Stoler et al., 1995*; *Buchwitz et al., 1999*; *Howman et al., 2000*; *Blower and Karpen, 2001*; *Talbert et al., 2002*). Moreover, the presence of CenH3 defines both canonical centromeres and neocentromeres in diverse organisms (*Dawson et al., 2007*; *Malik and Henikoff, 2009*). The presence of CenH3 homologs in all animals, fungi, and plants studied so far, together with their identification in distantly branching protist lineages has established the paradigm that CenH3-containing chromatin is an absolute requirement for centromere function (*Malik and Henikoff, 2003*; *Panchenko and Black, 2009*). In contrast to this prevailing paradigm, we show that multiple insect lineages have independently lost CenH3 despite preserving some other canonical kinetochore components. These CenH3 losses coincide with dramatic changes in centromere architecture, suggesting that alternate centromere configurations may render the CenH3 protein dispensable.

**eLife digest** Cell division is a fundamentally important process for living organisms. In eukaryotes, such as plants and animals, genomic DNA is tightly packaged into chromosomes, which needs to be copied and faithfully divided into daughter cells. Segregating the chromosomes is accomplished by the kinetochore, a protein complex that assembles on the chromosome and forms attachments to the machinery that provides the force for chromosome segregation. Kinetochores assemble on specialized chromosomal regions called centromeres. In most eukaryotes, kinetochore assembly relies on a centromeric protein called CenH3 that is essential for the process of chromosome segregation.

Most animal and plant species are monocentric—one part of the chromosome is dedicated to CenH3 loading and centromere function and paired chromosomes appear to be joined at a single point, or primary constriction. In contrast, holocentric species instead have centromeric activity distributed along the entire length of the paired chromosomes. How holocentricity arose from monocentricity over the course of evolution remains unclear.

Drinnenberg et al. took advantage of the fact that insects represent at least four independent transitions from monocentric to holocentric chromosomes. Several species of insects are holocentric—including butterflies and moths, bugs and lice, earwigs, and dragonflies—while others are monocentric—such as flies, bees, and beetles.

Drinnenberg et al. compared the repertoire of kinetochore proteins from each of these insect lineages and found that CenH3 was absent in all the holocentric insects examined but present in all the monocentric insect species. Despite the loss of CenH3 in the holocentric insects, they still had many of the kinetochore proteins—particularly, those proteins that attach to the machinery that forces chromosomes apart. Based on an evolutionary reconstruction, Drinnenberg et al. infer that each independent transition to holocentricity in insects likely introduced changes to the centromere that eliminated the need for the otherwise essential CenH3 protein.

This study challenges the notion that CenH3 is essential in all eukaryotes. Indeed, holocentric insects, which make-up 16% of the biodiversity of the currently known eukaryote species, appear to have evolved a completely novel way to define their centromeres, distinct from all the other eukaryotes.

Despite the universality of CenH3 in eukaryotes, centromeres are remarkably diverse (*Malik and Henikoff, 2009*). Most eukaryotic chromosomes are monocentric, that is, centromeres and kinetochore assembly are restricted to a defined chromosomal region. Monocentromeres can range dramatically in size, from 125 bp in budding yeasts to megabases in humans, and can be either genetically (sequence-dependent) or epigenetically defined. In contrast, holocentromeres have kinetochores attached along the extensive segments or even the entire length of chromosomes. First described by *Schrader (1935)*, holocentromeres have been best studied in the nematode *Caenorhabditis elegans* (*Dernburg, 2001*; *Maddox et al., 2004*; *Gassmann et al., 2012*; *Steiner and Henikoff, 2014*). However, holocentricity appears to have evolved independently in multiple eukaryotic lineages by convergent evolution (*Melters et al., 2012*).

## Results and discussion

To gain insight into kinetochore changes associated with transitions to holocentricity, we focused on insects in which holocentricity is believed to have evolved at least four times from monocentric ancestors (*Figure 1A*, *Figure 1—figure supplement 1*) (*Whiting, 2002*; *Grimaldi and Engel, 2005*; *Savard et al., 2006*) (modified from *Melters et al., 2012*). The strength of evidence about holocentricity varies among different insect orders; we briefly summarize this evidence in the 'Materials and methods' section. For instance, a strong consensus has emerged from many different species that both Lepidoptera and Hemiptera represent holocentric insect orders. In contrast, there is relatively modest evidence for holocentricity in Dermaptera, Odonata, and Phthiraptera. Nevertheless, the currently held consensus view is that each of the insect orders indicated in blue in *Figure 1A* is holocentric.

Using homology searches, we analyzed all the sequenced (monocentric and holocentric) insect genomes to identify protein components of the inner kinetochore complex. Initially, we focused on CenH3

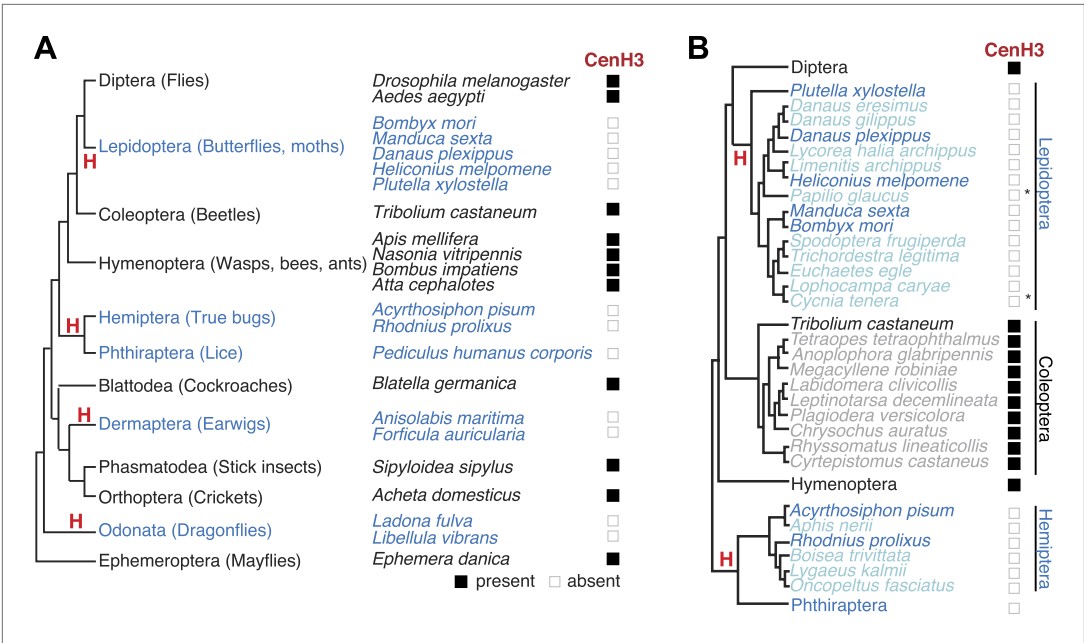

**Figure 1**. Holocentric insects lack CenH3. (**A**) Phylogeny of insect orders (species) examined in this study. Holocentric insect orders are indicated in blue, and inferred multiple transitions to holocentricity in insects are labeled with 'H'. Using protein homology searches of genomes or assembled transcriptomes, we inferred either the presence (black box) or absence (empty box) of CenH3. (**B**) CenH3 loss is widespread in Lepidoptera and Hemiptera, but not Coleoptera. Phylogenetic relationship of holocentric insects used for transcriptome assemblies (light blue), holocentric insects with sequenced genomes (blue), monocentric insects used for transcriptome assemblies (gray), and monocentric insects with sequenced genomes (black). The presence of contaminating microsporidian CenH3 transcripts is indicated with an asterisk.

The following source data and figure supplements are available for figure 1:

**Source data 1**. List of species with sequenced genomes and information to their corresponding analyzed proteomes.

**Source data 2**. Statistics of mRNA-Seq assemblies.

**Figure supplement 1**. Evolution of holocentricity in insects.

**Figure supplement 2**. Insect CenH3 comparison.

**Figure supplement 3**. Additional H3-like variants in holocentric insects.

**Figure supplement 4**. Fungal CenH3 transcript contaminants in the cockroach assembly.

**Figure supplement 5**. CenH3 transcript abundance in mRNA-Seq assemblies.

**Figure supplement 6**. CenH3 phylogeny in insects.

**Figure supplement 7**. CenH3 homologs in Coleoptera (beetles).

**Figure supplement 8**. Contaminating microsporidian CenH3 transcripts in the two lepidopteran assemblies.

**Figure supplement 9**. Chromosome spreads from various lepidopteran species generously provided by Frantisek Marec and Atsuo Yoshido.

that can be identified by its homology to histone H3, but distinguished from canonical H3 and other H3 histone variants via their distinct phylogenetic grouping and features such as a longer loop1 region and a highly divergent N-terminal tail (*Blower and Karpen, 2001*; *Malik and Henikoff, 2003*). These criteria have allowed the correct identification of CenH3 in all the eukaryotic genomes examined so far (*Malik and Henikoff, 2003*; *Talbert and Henikoff, 2010*). In addition to the well-studied CenH3 homologs in *Drosophila* species (*Henikoff et al., 2000*), we were able to identify CenH3 homologs in the monocentric insect orders: Diptera (flies and mosquitoes), Hymenoptera (wasps, bees, and ants), and Coleoptera (beetles) (*Figure 1A*, *Figure 1—figure supplement 2*). In contrast, we were unable to find CenH3 homologs in any of the five sequenced holocentric lepidopteran species (butterflies and moths) (*Figure 1A*). Extending our survey, we were similarly unable to find CenH3 in two additional insect orders previously reported to be holocentric: Hemiptera (true bugs) and Phthiraptera (lice). Together, these two orders represent an independent transition to holocentricity in insects (*Figure 1A*, 'Materials and methods') (*Whiting, 2002*). Although we found an additional H3-like gene in the hemipteran pea aphid *Acyrthosiphon pisum*, we found that it unambiguously phylogenetically clustered with canonical H3 proteins, rather than CenH3s (*Figure 1—figure supplement 3A,C*). Thus, CenH3 is missing from all sequenced genomes from eight holocentric insect species, but present in all sequenced monocentric insects.

We wanted to test this correlation between holocentricity and the absence of CenH3 in other mono- and holocentric insect orders beyond those with sequenced genomes. To search for CenH3, we carried out mRNA-sequencing (mRNA-Seq) analyses and transcriptome assemblies of five insects, and combined our analyses with transcriptome assemblies of three additional available insect mRNA-Seq data sets. Our analyses yielded assemblies from insects of four monocentric orders, *Blattodea germanica* (Blattodea, cockroaches), *Acheta domesticus* (Orthoptera, crickets), *Sipyloidea sipylus* (Phasmatodea, stick insects), and *Ephemera danica* (Ephemeroptera, mayflies) (https://www.hgsc.bcm.edu/). We were able to identify putative CenH3 homologs in the assemblies of all monocentric insects (*Figure 1A*, *Figure 1—figure supplement 2*). Identification of two bona fide CenH3 transcripts in the cockroach assembly led us to clearly attribute one of them to a fungal contamination via phylogenetic analyses (*Figure 1—figure supplement 4*) whereas the other was the cockroach CenH3 ortholog. We also obtained transcriptomes of three additional insects from two additional insect orders previously reported to be holocentric: *Anisolabis maritima* (Dermaptera, earwigs), *Libellula vibrans* and *Ladona fulva* (Odonata, dragonflies and damselflies) (https://www.hgsc.bcm.edu/). Additionally, we included a transcriptome assembly from *Forficula auricularia* (Dermaptera, earwigs) that had recently become available (*Roulin et al., 2014*). Together with Lepidoptera and Hemiptera/Phthiraptera, these new holocentric insect orders (Dermaptera, Odonata) represent all independent transitions to holocentricity in insects (*Figure 1A*, 'Materials and methods'). In contrast to the monocentric insects, we could not detect CenH3 in the assemblies derived from any of the holocentric insects. Although we did obtain an H3-like gene in the *L. vibrans* assembly, evolutionary analyses support its phylogenetic grouping with H3 rather than CenH3 (*Figure 1—figure supplement 3B,C*).

We considered the possibilities that either insufficient coverage of the transcriptome assembly and/or low expression of CenH3 led to our inability to detect CenH3 transcripts. To address the first possibility, we compared the probabilities of finding a CenH3 homolog between mono- and holocentric assemblies. Using the well-characterized proteome from *Tribolium castaneum* as a benchmark, we tested if similar numbers of proteins could be predicted in our assemblies (*Figure 1—source data 2*). Indeed, our analyses revealed comparable or even higher number of predicted proteins in the holocentric compared to the monocentric assemblies, implying at least equivalent transcriptome coverages of the holocentric and monocentric assemblies. Second, we found that CenH3 transcripts are not rare in monocentric insects; CenH3 transcript abundance was at least at the 50th percentile in all instances except the cockroach assembly (*Figure 1—figure supplement 5*). We therefore conclude that CenH3 is likely absent in the four holocentric odonatan and dermapteran insects examined as it is absent in the genome sequences of the holocentric lepidopteran and hemipteran/phthirapteran insects. Thus, we conclude that each of the four independent transitions to holocentricity in insects was associated with CenH3 loss.

Phylogenetic analyses of CenH3 and other H3 proteins based on their homologous histone-fold domains reveal a topology of insect CenH3s that is largely consistent with the expected branching order of the insect species (*Whiting, 2002*; *Grimaldi and Engel, 2005*; *Savard et al., 2006*) (*Figure 1—figure supplement 6*), confirming that the absence of CenH3 in holocentric insects is due to recurrent loss rather than reinvention or horizontal transfer of CenH3 in monocentric insect lineages.

To precisely date when CenH3 loss occurred in holocentric insects, we took advantage of additional transcriptome assemblies of mRNA-Seq data sets from nine coleopteran, nine lepidopteran, and four hemipteran species (*Negre et al., 2006*; *Zhen et al., 2012*) (https://www.hgsc.bcm.edu/). These additional assemblies confirmed not only CenH3 presence in all (monocentric) coleopteran species (*Figure 1—figure supplement 7*), but also confirmed its absence in all hemipteran and lepidopteran species. Although we obtained CenH3-like sequences in two lepidopteran assemblies, we could unambiguously attribute them to microsporidian contaminants via phylogenetic analyses (*Figure 1—figure supplement 8A,B*). As a result, we estimate that CenH3 loss probably preceded or occurred close to the emergence of these insect orders, at least 120 million years ago in Lepidoptera (*Hedges et al., 2006*) and likely 300 million years ago in the common ancestor of Hemiptera and Phthiraptera (*Grimaldi and Engel, 2005*).

We next investigated the genome and transcriptome data sets to ask whether the onset of holocentricity also correlated with the loss of other inner kinetochore components, or CCAN (constitutive centromere associated network) components as previously defined in vertebrates and fungi (*Hori et al., 2008*; *Westermann and Schleiffer, 2013*). We identified CenpI, CenpL/M/N, and the DNA-proximal CenpS/X proteins. We note that since the CenpT/W proteins, which are obligatory for the CenpS/X kinetochore function, are absent in all insects, it is likely that the CenpS/X complex serves roles in DNA damage (*Singh et al., 2010*) rather than at the kinetochore. Nonetheless, several inner kinetochore components continue to be present even in the CenH3-deficient species (*Figure 2A*). This is somewhat unexpected since the recruitment of these components has been shown to depend on CenH3 in animals and fungi (*Westhorpe and Straight, 2013*). It will therefore be important in future to test if these kinetochore components localize to centromeres in the absence of CenH3; this would imply that holocentric insects have adopted a CenH3-independent inner kinetochore assembly pathway.

We did find the inner kinetochore protein CenpC to be absent from all insects that lack CenH3 (*Figure 2A*). CenpC is the direct DNA-binding partner of CenH3 and together with CenH3 constitutes the centromere core components (*Carroll et al., 2010*). Although CenpC homologs are not as easily identifiable as CenH3, two motifs are conserved enough to allow cross-species homology searches. Of these, the CenpC motif is universally found in CenpC proteins and has been shown to interact with the CenH3 C-terminus (*Kato et al., 2013*). In addition, most animal and fungal CenpC proteins contain a cupin fold domain (Pfam PF00190), about 100 amino acids downstream of the CenpC motif (*Talbert et al., 2004*). Although cupin domains are not unique to CenpC proteins, CenpC-borne cupin domains can be easily distinguished from non-CenpC cupin domains based on phylogeny (*Dunwell et al., 2001*). Homology searches using the CenpC motif revealed high-scoring CenpC-like motifs in most monocentric insects (HHpred p value range $2 \times 10^{-9}$ to $4.6 \times 10^{-22}$) except in the *Tribolium castaneum* (genome) and cricket (transcriptome) (*Figure 2A*). We found that CenpC proteins from mayflies and stick insects (both monocentrics) possess both the CenpC motif and the associated cupin domain (*Figure 2C*). In contrast, putative CenpC proteins from Hymenoptera and cockroach (also monocentric) contained the CenpC motif but not the cupin domain. We were unable to find the CenpC motif in any of the holocentric species (genomes or transcriptomes). Intriguingly, although holocentric dragonflies encode a protein with a clearly identifiable CenpC cupin domain (*Figure 2D*, *Figure 2—figure supplement 1*), its CenpC motif has decayed (*Figure 2C*). We speculate that since the CenpC motif interacts with CenH3, CenH3 loss in dragonflies may have allowed the degeneration of its CenpC motif. The absence of CenH3-interacting CenpC motifs in dragonflies and other holocentric insects also serves as an independent confirmation for the losses of CenH3 in these lineages.

In contrast to the variation in inner kinetochore protein repertoires, homologs of outer kinetochore components including Ndc80 and Mis12 are almost universally conserved in most insect orders (*Figure 2A*). This suggests that although insects vary widely in the DNA-proximal part of their kinetochore, they have largely conserved the means to attach the kinetochore to the spindle apparatus. This further implies that holocentric insects likely employ CenH3-independent ways of connecting centromeric DNA to the rest of a relatively conserved kinetochore complex. Our findings in holocentric insects contrast with the recent findings of CenH3 absence in kinetoplastids, which not only lack CenH3 but also possess an entirely different repertoire of kinetochore proteins (*Akiyoshi and Gull, 2013*, *2014*). Our findings are more reminiscent of male meiosis in holocentric *C. elegans*, in which the outer kinetochore components appear to localize independent of CenH3 (*Monen et al., 2005*); CenH3

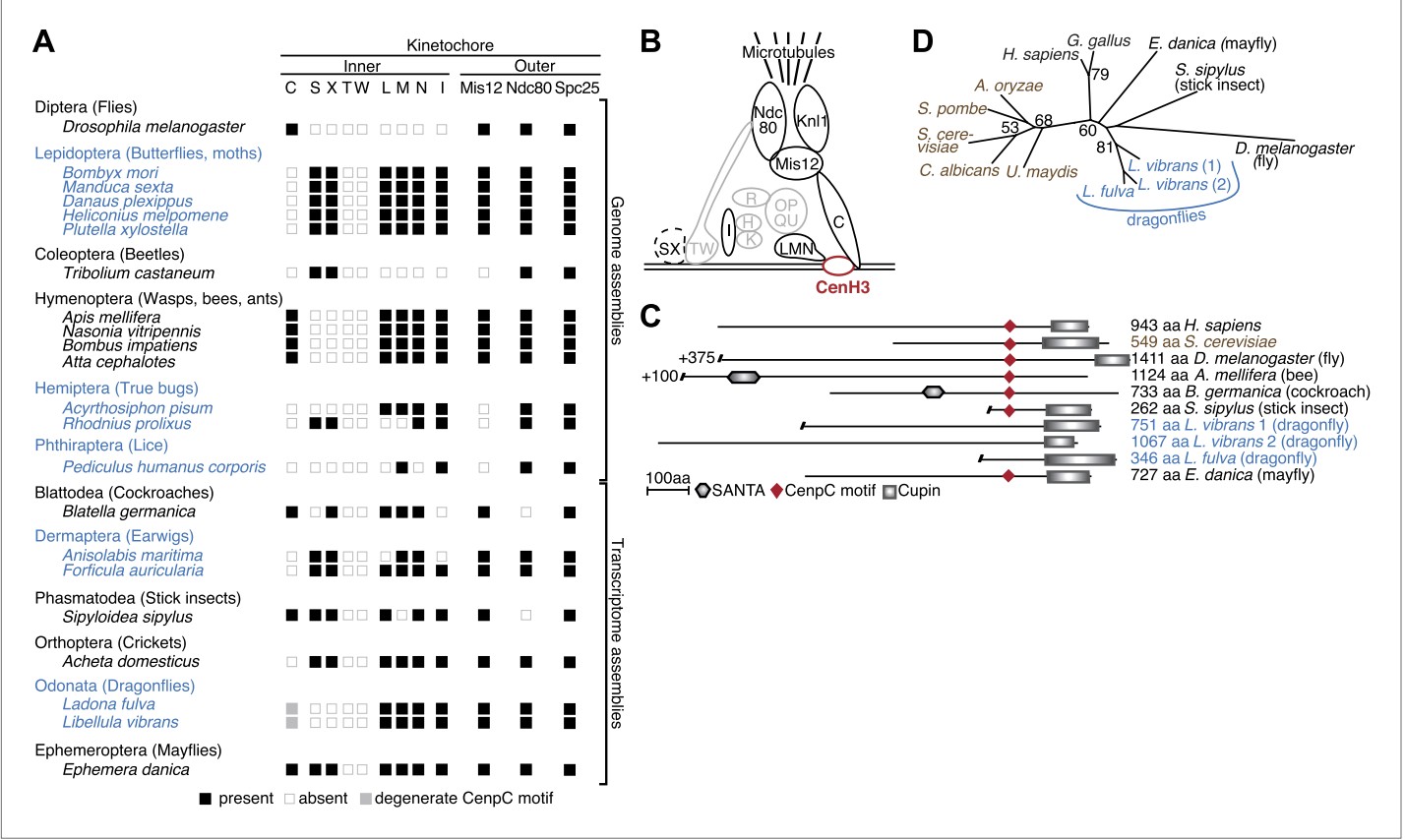

**Figure 2**. Evolution of the kinetochore composition in insects. (**A**) Kinetochore proteins in insects. Using protein homology searches of the genome or assembled transcriptomes of monocentric (indicated in black) and holocentric insects (indicated in blue), we inferred either the presence (black box) or absence (empty box) of inner and outer kinetochore protein components. The presence of a putative CenpC homolog without a recognizable CenpC motif in Odonata is indicated by a filled gray box. We also found a weak match to a putative Mis12 homolog in *T. castaneum* (TC001997); however its weak homology relative to other insect Mis12 proteins leads us to assign this to be only a tentative match. (**B**) Schematic structure of a kinetochore largely based on its characterization in vertebrates and fungi (***Hori et al., 2008***; ***Westermann and Schleiffer, 2013***). Components found in insects are highlighted in black, whereas components that are absent in all insects examined are in gray. The dashed line around the CenpS/X complex indicates that its kinetochore localization is unlikely in insects. (**C**) Putative CenpC proteins in insects. Schematics of structural domains and the CenpC motif of human, yeast, and insect CenpC proteins are shown. (**D**) Maximum likelihood tree of cupin domains of dragonfly (blue), fungal (brown), and other animal (black) CenpC proteins. Bootstrap percentages above 50 are indicated.

The following source data and figure supplements are available for figure 2:

**Source data 1**. Accession numbers or sequences for all insect kinetochore proteins analyzed or described in this study.
**Figure supplement 1**. CenpC-Cupin domain alignment.

is nevertheless essential for *C. elegans* mitosis (***Buchwitz et al., 1999***). Thus, holocentric insects, which account for 16% of all named eukaryotic species (***Grimaldi and Engel, 2005***), appear to be unique among complex eukaryotes in entirely losing their dependence on CenH3s.

## Concluding comments

Given the essential requirement of CenH3 for chromosome segregation in monocentric contexts as shown in *Drosophila melanogaster* (***Blower and Karpen, 2001***), we hypothesize that independent transitions to holocentricity preceded the losses of CenH3s in insects. However, since other holocentric animals (including other arthropods) and plants still encode for CenH3 (***Buchwitz et al., 1999***; ***Heckmann et al., 2011***) (***Figure 2—source data 1***), we speculate that another, distinct event allowing CenH3 loss in holocentric insects must have occurred early in insect evolution, in the common ancestor of dragonflies and flies (***Figure 3***). Such a determinant may have been the lineage-specific evolutionary

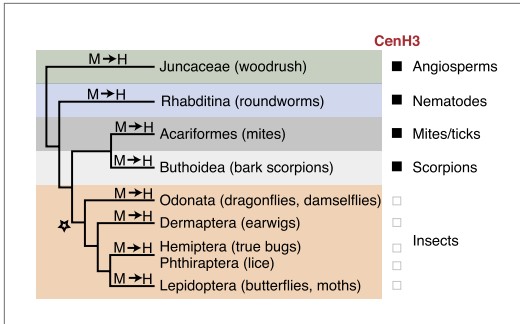

**Figure 3**. CenH3 losses associated with holocentric transitions are unique to insects. Phylogenetic relationship of holocentric lineages are schematized. Transitions to holocentricity (M → H) and the absence (black box) or presence (empty box) of CenH3 are indicated. Inferred evolutionary origin of a first 'potentiating event' (indicated with a star) together with subsequent recurrent transitions to holocentricity allowing the loss of CenH3 in four insect lineages.

invention of a centromeric protein (**Ross et al., 2013**), which may have conferred these independently derived holocentric lineages with a unique ability to carry out mitosis in a CenH3-independent manner, thereby relaxing the selective pressure to maintain CenH3 in the genome. Thus, changes in centromere architecture may have rendered dispensable one of the most defining proteins associated with centromere function in almost all eukaryotes.

## Materials and methods

### Evidence for holocentric insect orders

Four main criteria have been used to diagnose holocentricity by studying the mitotic behavior of chromosomes. These include (1) the absence of a primary constriction in metaphase chromosomes, (2) parallel migration of sister chromatids in mitotic anaphase, (3) persistence of chromosomal fragments and/or low rate of lethality upon X-ray irradiation, and (4) the presence of kinetochore plates covering large surfaces of each chromatid during mitosis. In addition to these four primary mitotic criteria for holocentricity, we consider certain properties at meiosis suggestive of holocentricity. Even though mitosis of holocentric chromosomes is straightforward, holocentric chromosomes would face kinetochore geometry problems during meiosis. Therefore, meiosis in holocentric species requires unique adaptations and can therefore be used as indications for holocentricity. Such adaptations include 'inverted meiosis' (i.e., inverting the order of the two meiotic divisions) or the change of kinetic activity on chromosomes from one region to another between meiotic divisions (**Melters et al., 2012**).

### Lepidoptera (butterflies, moths)

Most Lepidoptera have numerous, small chromosomes, challenging conclusions about the nature of centromeres purely on the basis of light microscopy. Nonetheless, convincing evidence for holocentric chromosomes exists in *Bombyx mori*, one of the most well-studied holocentric insects (**Murakami and Imai, 1974**). *B. mori* chromosomes are rod- and dot-shaped and lack primary constrictions. Also, during anaphase, sister chromatids migrate in parallel. Furthermore, chromosomal fragments induced by X-ray irradiation behave normally through a number of cell generations (**Murakami and Imai, 1974**). More recently, immuno-fluorescence studies showed the localization of *B. mori* chromosomal passenger complex protein INCENP along the whole length of mitotic chromosome (**Mon et al., 2014**).

Studies from additional lepidopteran insects support the general conclusion of holocentricity in all members of Lepidoptera. Mitotic chromosomes of *Orthosia gracilis* appear to lack a primary constriction (**Traut and Mosbacher, 1968**). Electron microscopy (EM) in *Origya thyellina* and *Origya antiqua* chromosomes report spindle microtubule attachments to large kinetochore plates, covering more than 70% of the entire surface of the chromosomes (**Wolf, 1994**; **Wolf et al., 1997**). This has led to the proposal of a polykinetic organization of *Origya* chromosomes. Original images of mitotic chromosomes of *Samia cynthia* (unpublished), *Ectomyelois ceratoniae* (**Mediouni et al., 2004**), and *Cydia pomonella* (**Fukova et al., 2007**), as well as an image of metaphase I chromosomes of *Ephestia kuehniella* (unpublished) (**Figure 1—figure supplement 9**), generously provided by Frantisek Marec and Atsuo Yoshido, show that the chromosomes of these species lack primary constrictions. *E. kuehniella* chromosomes have also been subject to EM studies that have reported large kinetochore plates covering the chromosomal surface. However, compared to *Origya* species, the extent of kinetochore occupancy in *E. kuehniella* was estimated to be not as wide. Therefore, the kinetic organization of *E. kuehniella* chromosomes was interpreted to represent an intermediate between monocentricity and complete holocentricity (**Wolf, 1994**). Finally, the presence of holocentric chromosomes is also consistent with

experimental data reporting high resistance to radiation in multiple lepidopteran species (*LaChance and Graham, 1984*; *Koval, 1996*; *Marec et al., 2001*).

In contrast to the general conclusion about holocentricity in Lepidoptera, two lepidopteran species, *Pieris brassica* and *Polyommatus icarus*, have been reported to be monocentric in mitosis inferred by the presence of primary chromosomal constrictions observed by light microscopy (*Bigger, 1975*; *Rishi and Rishi, 1978*). However, meiotic chromosomes from one of these species were reported to lack a localized centromere (*Bigger, 1975*). Moreover, due to low zygotic lethality and frequent presence of chromosomal translocations after X-ray irradiation in *Pieris brassicae*, it has been proposed that *P. brassicae* chromosomes are in fact holocentric (*Bauer, 1967*).

Taken together the cytogenetic results on Lepidoptera, Bauer proposed that '*sufficient proof exists to conclude that all Lepidoptera have holokinetic chromosomes*' (*Bauer, 1967*). We therefore favor this general conclusion that Lepidoptera are holocentric.

## Trichoptera (caddis flies)

Trichoptera are a sister lineage to the Lepidoptera. An early study suggested the possibility of holocentric chromosomes in the trichopteran species *Limnophilus borealis* and *Limnophilus decipiens* (*Suomalainen, 1966*). More recent EM data of *L. decipiens* instead proposed an intermediate form between the holo- and monocentric type (*Wolf et al., 1997*). Additional trichopteran species have not been examined. Thus, although the consensus view remains that members of Trichoptera are holo- rather than monocentric, this conclusion is not as well supported as for some of the other insect orders.

As a result, it remains unknown whether the transition to holocentricity occurred in the common ancestor of all Lepidoptera or in the common ancestor of Lepidoptera and Trichoptera. We have not assessed the status of *CenH3* genes in Trichoptera.

## Hemiptera (true bugs)

Similar to Lepidoptera, there are numerous reports of holocentric chromosomes in several species of Hemiptera (*Melters et al., 2012*). Chromosomes of multiple hemipteran species belonging to the hemipteran suborder Heteroptera (*Oncopeltus fasciatus* [*Wolfe and John, 1965*], *Euschistus servus*, *Euschistus tristigmus*, and *Sotubea pugnax* [*Hughes-Schrader and Schrader, 1961*]), as well as from the suborder Sternorrhyncha (aphids [*Sun and Robinson, 1965*; *Mandrioli et al., 2011*]) lack primary constrictions. EM studies in *O. fasciatus* (*Comings and Okada, 1972*) and *Rhodnius prolixus* (*Buck, 1967*) reporting kinetochore occupancy covering 75% and 100% of mitotic chromosomes respectively, further support the presence of holocentric chromosomes in Heteroptera. Experimental support for holocentricity was also obtained using chromosomal fragmentation studies, which demonstrated that fragments induced by X-rays are able to propagate themselves mitotically through many cell generations (*Hughes-Schrader and Schrader, 1961*; *Khuda-Bukhsh and Datta, 1981*; *Kuznetsova and Sapunov, 1985*). Furthermore, C-banding of several aphid species revealed that heterochromatic areas are interspersed with euchromatic areas whereas heterochromatin of monocentric chromosomes is concentrated in one part of the chromosome (reviewed in *Manicardi et al., 2002*).

There also exists ample evidence of holocentricity in Hemiptera based on meiotic observations that reveals inverted meiosis of sex chromosomes (*Schrader, 1935*; *Viera et al., 2009*) and/or change of kinetic activity on autosomes from one region to another between meiosis I and II (*Perez et al., 1997*).

As a result of all these studies, we agree with the previous conclusion that '*the diffuse nature of the hemipteran kinetochore is attested by a wealth of observational evidence*' (*Hughes-Schrader and Schrader, 1961*).

## Phthiraptera (lice)

Phthiraptera is a sister order with Hemiptera. However, compared to Hemiptera, there is relatively little data characterizing the nature of the centromere in this order. Chromosome structure has been investigated by cytological analysis in *Haematopinus suis* and *Menacanthus stramineus* (*Tombesi and Papeschi, 1993*) and in *Bovicola limbata* and *Bovicola caprae* (*Tombesi et al., 1999*). The presence of holocentric chromosomes was reported based on the cytological observations, including the lack of primary constrictions and parallel sister chromatid migration during anaphase. Experimental support has been obtained in *H. suis* by Bayreuther (*Bayreuther, 1955*), who observed a regular behavior of chromosome fragments during cell division after irradiating first and second instar individuals.

Although Hindle and Pontecorvo reported the existence of primary constrictions in *Pediculus corporis* chromosomes (without providing cytological data) indicating the presence of monocentric

chromosomes (*Hindle and Pontecorvo, 1942*), this view was challenged by Bayreuther, who instead proposed holocentric chromosomes in *Pediculus*, as this is more consistent with observations following DNA fragmentation experiments (*Pontecorvo, 1944*), including the lack of harmful effects on *Pediculus* spermatozoa and the absence of reports about akinetic fragments or mitotic chromosome bridges (*Bayreuther, 1955*). To our knowledge no photographic evidence exists showing the presence of monocentric chromosomes in Phthiraptera. Though the listed photographic support leading to the author's interpretation of holocentricity is difficult to assess today, we favor the consensus interpretation that Phthiraptera are holocentric, particularly due to the strength of evidence based on studies assessing chromosomal inheritance following irradiation.

Thus, the common ancestor of Hemiptera and Phthiraptera may have undergone a single transition to holocentricity.

## Dermaptera (earwigs)

Multiple species of Dermaptera have been reported to be holocentric. Ortiz studied chromosome cytology of seven species of Dermaptera by microscopy and reported that '*the chromosomes show no localized centromere, as inferred from their structure, mode of division and anaphase movement*' (*Ortiz, 1969*). In this study, the most conclusive data are derived from *Labidura ripura*, a close relative to *Forficula auricularia* belonging to the same suborder. *L. ripura* has the smallest number of chromosomes with the largest relative length facilitating the assessment of lack of primary constrictions (*Ortiz, 1969*). Furthermore, the chromosomes of two additional earwigs have been examined. In one study in *Labidura truncata,* Webb reported that '*the primary constrictions of fixed centromeres do not show, and uninterrupted chromosomes, characteristic of holocentric chromosomes…*' (*Webb, 2009*). In another study, MJD White examined *Hemimerus bouvieri* (*White, 1971*) and observed rod-shaped chromosomes with no constriction and parallel migration of sister chromatids during anaphase.

Two studies from the earwig *F. auricularia* challenged the consensus view that Dermaptera are holocentric (*Callan, 1941*; *Henderson, 1970*). Both Hendersen and Callan reported the presence of primary constrictions in *F. auricularia* chromosomes. The view of these authors, however, has been challenged by Ortiz who found '*chromosomal spindle fibers exist along the mitotic chromosomes*' thus proposing that '*the chromosomes of Dermaptera have a diffuse centromere*' (*Ortiz, 1969*). We also favor the more general conclusion that Dermaptera are indeed holocentric.

## Odonata (dragonflies)

The nature of centromeres in Odonata has been a subject of great debate. Early studies concluded that chromosomes in this order are metacentric (*Oksala, 1943*), acrocentric (*Chauduri and Das Gupta, 1949*; *Seschachar and Bagga, 1962*), or holocentric (*Cumming, 1964*; *Kiauta, 1969a*, *1969b*). The most recent study on odonatan chromosomes, Nokkala et al. applied silver staining methods to visualize metaphase chromosomes and the behavior of male meiotic chromosomes in two dragonfly species (*Nokkala et al., 2002*). This study clearly demonstrated the absence of a primary constriction and parallel alignment of metaphase chromosomes. Furthermore, this study also showed that during meiosis different chromosomal regions (telomeres in the first division and the middle parts of chromosomes in the second) showed kinetic activity. At present, although the two species studied most recently appear to be holocentric, we cannot conclude that this is representative of the entire order. Indeed, it is formally possible that holocentricity might have arisen within Odonata rather than in the common ancestor. Although the centromere status of the species that we sequenced has not been determined, they belong to the same suborder as the holocentric dragonflies studied by Nokkala.

## Zoraptera (zorapterans)

Zoraptera represent an insect order that is relatively poorly studied especially with respect to their chromosome structure. Although there is also a significant uncertainty of their phylogenetic position, they may represent a sister lineage to the Dermaptera. In investigations of male meiotic chromosomes in *Zorotypus hubbardi*, no primary constrictions were observed at any meiotic stage (*Kuznetsova et al., 2002*) leading to the conclusion that they are holocentric. We have been unable to obtain samples of Zoraptera to assess the status of their *CenH3* genes.

## Ephemeroptera (mayflies)

Ephemeropteran insects have been previously referred to as being holocentric (*Melters et al., 2012*). However, cytogenetic studies of many species of this order (including *E. danica* that we included in our

transcriptome analyses) have clearly reported that the centromeres are metacentric or acrocentric (*Kiauta and Mol, 1977*; *Soldan and Putz, 2000*). Kiauta and Mol compared the cytological evidence for monocentric chromosomes in Ephemeroptera directly to their cytological evidence of holocentric chromosomes in species belonging to the sister order Odonata (*Kiauta, 1969a*, *1969b*). As a result, the authors conclude that '*any cytogenetic similarities between the two orders are completely lacking*' (*Kiauta and Mol, 1977*). We therefore conclude that the assignment of holocentricity to the Ephemeroptera is incorrect and they should be assigned a monocentric status instead.

## RNA isolation and sample preparation

We obtained German cockroaches (*Blattella germanica*), crickets (*Acheta domesticus*), and dragonfly nymphs (*Libellula vibrans*) from Carolina Biological Supply Company. Earwigs (*Anisolabis maritima*) and stick insects (*Sipyloidea sipylus*) were collected from the field and obtained from captive populations, respectively. For each insect, we dissected the head and thorax (to minimize the contamination by gut microbiota), which were then homogenized and total RNA was isolated using the TRIZOL reagent according to the manufacturer's instructions (Invitrogen, Carlsbad, CA). RNA was treated with DNaseI (Ambion) to remove DNA contamination and purified using Qiagen RNeasy Mini Kit. mRNA-sequencing (mRNA-Seq) libraries were prepared using Illumina's mRNA-Seq Sample Prep Kit (Illumina, San Diego, CA) following the manufacturer's protocol. Barcoded libraries were multiplexed and sequenced as 50-bp paired-end reads on the HiSeq 2000 platform. Reads were parsed by barcodes, and the number of reads for each species is listed in *Figure 1—source data 2*.

### Assemblies of mRNA-Seq reads

We carried out de novo assembly of mRNA-Seq reads using a combination of Velvet (*Zerbino and Birney, 2008*) and Oases (*Schulz et al., 2012*) programs. We generated assemblies of the publically available 100-bp paired-end mRNA-Seq libraries from mayfly (*Ephemera danica*) and scarce chaser (*Ladona fulva*) (https://www.hgsc.bcm.edu/), as well as our own 50 bp paired-end mRNA-Seq libraries, using a k-mer length of 21. K-mer lengths were set to 31 for additional assemblies of publically available 100 bp single-end or paired-end mRNA-Seq data sets from several lepidopteran, coleopteran, and hemipteran species (*Figure 1B* [*Zhen et al., 2012*]), as well as from *Drosophila melanogaster* strain Oregon-R ovaries (SRR384962, http://www.ncbi.nlm.nih.gov/sra/SRX109311). We generated additional assemblies for the cockroach (*B germanica*), dragonfly (*L. vibrans*), earwig (*A. maritima*), and cricket (*A. domesticus*) libraries with k-mer lengths set to 17, 23, and 33 using the SPAdes assembler (*Nurk et al., 2013*). A transciptome assembly of a second dermapteran mRNA-Seq data set from the earwig *Forficula auricularia* (*Roulin et al., 2014*) (SRR1043671, SRR1048074, SRR1051467) was also included in our analyses.

## Comparing transcriptome coverages in mRNA-Seq assemblies

Transcripts at least 250 nucleotides in length were selected. In cases where multiple isoforms were present, we chose the longest isoform. We used the *Tribolium castaneum* proteome (16,644 proteins, version as of December 2013) and the *D. melanogaster* proteome (30,305 proteins, version as of May 2014) to query the assembled transcripts using BLAST (tblastn). Significant alignments (*E* value < $10^{-10}$) were counted considering only one alignment per transcript of *T. castaneum*/*D. melanogaster* protein. The results are listed in *Figure 1—source data 2*.

We predicted similar fractions of *T. castaneum* (and *D. melanogaster*) proteins in each of our assemblies from mono- or holocentric species implying comparable transcriptome coverages. Overall, we could predict 24–48% of the *T. castaneum* proteome in assemblies from monocentric insects. Each of these assemblies also contained CenH3. As expected due to their closer evolutionary relationship compared to other insect orders, the biggest *T. castaneum* proteome fractions were predicted in assemblies generated from beetle mRNA-Seq data sets. Protein predictions in assemblies from holocentric insects yield fractions ranging from 25% to 44% (*Figure 1—source data 2*). Proteins that were not found either did not exist in the corresponding species or were too divergent to pass our alignment threshold (*E* value <$10^{-10}$), or were not expressed in the sample (or tissues analyzed). The assembly of *D. melanogaster*, an insect species with an annotated proteome, was used to assess the expected fraction of predicted *T. castaneum* proteins. 28% of all *T. castaneum* proteins were predicted (total 16,644 proteins) in the *D. melanogaster* assembly, which falls within the range of predicted fractions in other assemblies. This assembly covers 21% of all *D. melanogaster* proteins (total 30,305 proteins) and includes the *D. melanogaster* CenH3 homolog.

## Estimating transcript abundance in mRNA-Seq assemblies

The assembled transcripts were parsed into a non-redundant set selecting the longest isoform per transcript. We mapped the mRNA-Seq reads to the assembled set of transcripts using Bowtie 2 using default parameters (*Langmead and Salzberg, 2012*). Mapped reads were normalized to transcript lengths to estimate transcript abundances. Transcripts (longer than 200 nt) were rank-ordered based on their abundances. We determined the ranked percentiles corresponding to transcripts encoding for CenH3 homologs and kinetochore proteins, where low percentiles correspond to low abundances, while high percentiles correspond to high abundances (*Figure 1—figure supplement 5*). Although CenH3 was only at the bottom 10th percentile in terms of abundance in the cockroach transcriptome assembly, CenH3 transcript abundance was in the top 50th percentile or better in the mayfly, the stick insect, the cricket, and the *D. melanogaster* assemblies. Thus, the absence of CenH3 in the assemblies of all holocentric insect species is likely not due to low CenH3 expression levels.

## Searches for CenH3 and kinetochore proteins

### CenH3 homology search in insects with sequenced genomes

NCBI non-redundant protein databases or specialized proteomes (*Figure 1—source data 1*) and whole genomes were searched using BLAST (psi-blast, blastp, and tblastn). We used *D. melanogaster* histone H3 (canonical histone H3 sequence is almost invariant among insects) and *D. melanogaster* CenH3 homolog Cid (FBgn0040477) as queries for our initial searches. Newly obtained CenH3 homologs were subsequently added to the list of query proteins, in iterative searches. We tested any putative homolog thus obtained for histone folds by HHpred (*Soding et al., 2005*) and for diagnostic features of known CenH3 homologs (*Malik and Henikoff, 2003*) including an extended loop one region, absence of specific amino acids including glutamine, phenylalanine, and threonine at positions 69, 85, and 118, respectively (as compared to canonical H3). These criteria have allowed the successful identification of CenH3 homologs in animals, plants, and even in distantly related protozoa (*Malik and Henikoff, 2003*), while revealing the absence of CenH3 in kinetoplastids (*Talbert and Henikoff, 2010*) that has recently been confirmed (*Akiyoshi and Gull, 2014*). In addition to those features, phylogenetic analyses support homology of the newly identified CenH3s to other known CenH3 proteins (*Figure 1—figure supplement 6*). All identified homologous proteins or protein fragments are listed in *Figure 2—source data 1*.

In addition to canonical H3, we found the universally conserved H3 variant H3.3 that is almost identical to H3 in all genomes of mono- and holocentric species. All genomes examined from monocentric insects contained an additional H3-like protein with blastp *E* values in the order of $10^{-10}$, which corresponds to CenH3. In contrast, apart from H3 and H3.3, no additional functional H3-like protein could be detected in any of the genomes of holocentric insects except for a putatively nonfunctional H3-derived variant in the *Acyrthosiphon pisum* genome (*Figure 1—figure supplement 3A,C*).

Although a putative *B. mori* CenH3 homolog with 23% identity to the histone-fold domain of the *Caenorhabditis elegans* CenH3 homolog (hcp-3) was described in the *B. mori* genome sequence publication (*Xia et al., 2004*), we were unable to replicate these findings; our analyses revealed no CenH3 in the *B. mori* genome, consistent with a more recent study (*Mon et al., 2011*) that also was unable to identify a CenH3 homolog in *B. mori* (however no details were provided about the prediction criteria and results leading to this conclusion).

### CenH3 searches in transcriptome assemblies and expressed sequence tag (EST) databases

We surveyed transcriptome assemblies and EST databases via tblastn searches using canonical H3 and any identified insect CenH3 homolog as query proteins. As expected, those searches revealed the presence of the universally conserved H3 variant H3.3 in all assemblies. In addition to H3.3, assemblies from all monocentric species contained another H3-like protein corresponding to their CenH3 homologs. CenH3 encoding transcript fragments in the monocentric coleopteran *Tetraopes tetraophthalmus* and *Megacyllene robinae* assemblies were detected in the raw 100 bp reads using the *Anoplophora glabripennis* CenH3 as query. Assemblies or EST data (*Negre et al., 2006*) from holocentric insects did not contain any additional insect H3-like hits apart from H3.3 and a recent H3-derived variant in the *L. vibrans* assembly (*Figure 1—figure supplement 3B,C*). Two lepidopteran assemblies revealed CenH3 encoding transcripts that could easily be attributed to microsporidan contamination

via phylogenetic analyses (*Figure 1—figure supplement 8A,B*). To rule out the possibility that CenH3 encoding read fragments are present in the mRNA-Seq libraries but not assembled to transcripts, raw 100 bp reads from any of the Lepidoptera as well as from *L. fulva* library were searched for H3-like protein fragments using BLAST (tblastn). Apart from H3 or H3.3 encoding read fragments, no additional CenH3 encoding read fragments could be identified in those libraries, whereas the same search yields significant hits (with *E* values down to $5 \times 10^{-9}$) to the CenH3 homolog in the monocentric mayfly mRNA-Seq data set. The 50 bp reads of the other insect libraries were not suitable for these analyses because the translated protein fragment is too short to yield significant alignments to the CenH3 homologous fragments using H3-like query proteins. Consistent with our analyses based on the Velvet/Oases assemblies, searches for CenH3 homologs using BLAST (tblastn) in the SPAdes assemblies from two holocentric species (earwig and dragonfly) did also not reveal any hits apart from H3.3.

The transcriptome assembly of the second earwig, *F. auricularia*, was recently generated using a combination of Roche 454 reads, Illumina 100 bp and 150 bp paired-end reads (*Roulin et al., 2014*). We searched H3-like proteins in the *F. auricularia* database using the associated BLAST server (http://gdcws1.ethz.ch/blastdb_fau/blast.html) and in the transcripts using tblastn. No CenH3 homolog was found.

## CenpC homology search

Transcripts, genomes, or annotated proteomes were searched (psi-blast, blastp, and tblastn) as described above using CenpC motifs, CenpC-C-terminal regions including the cupin domain or full-length CenpC proteins of human, yeast, and insect homologs. Newly identified CenpC homologs were added to the list of query proteins for iterative searches. A recent study searched selected insect genomes for CenpC homologs and other kinetochore proteins, without providing sequence information or accession numbers (*Schleiffer et al., 2012*). A CenpC homolog was reported in the beetle *T. castaneum*, while CenpC was not found in several species of Hymenoptera. Our searches did not reveal a recognizable CenpC homolog in *T. castaneum*; however, we identified several putative hymenopteran CenpC homologs.

The assembled *L. fulva* (scarce chaser) CenpC fragment only covered the 3′ end of the mRNA transcript including the stop codon of the open reading frame. The 5′ end of the fragment was manually extended with unassembled read fragments of the *L. fulva* library, identified by sequence similarity (tblastn) to the homologous *L. vibrans* CenpC protein (SPAdes Node 13,580).

Cupin domains are found in many protein subfamilies, which form distinct clades in phylogenetic analyses (*Dunwell et al., 2001*). Reciprocal best hit analyses using the dragonfly cupin domains identified CenpC-like cupin domains of various eukaryotic CenpC homologs, therefore confirming the homology of the dragonfly cupins to CenpC-like cupin subfamily.

## 5′ RACE analyses

The dragonfly CenpC fragment (SPAdes node 13,580) identified in the assemblies was extended towards the 5′ end of the transcript by 5′ RACE analyses using Invitrogen's 5′RACE System for Rapid Amplification of cDNA Ends. 10 μg total RNA was treated with DNaseI (Ambion) to remove DNA contamination. Gene-specific primers complementary to the assembled CenpC fragment were used for first strand cDNA syntheses using Superscript III (Invitrogen, Carlsbad, CA) following manufacturer's instructions. The cDNA was treated with RNase H and purified using S.N.A.P. columns (Invitrogen, Carlsbad, CA). Half of the purified cDNA was used for terminal deoxynucleotidyl transferase tailing reaction with dCTP. Tailed cDNA was precipitated and used for final amplification with gene-specific primers and 5′ RACE abridged anchor primer provided in the kit.

## Kinetochore protein homology searches

While most inner kinetochore components found in vertebrates and fungi are absent in *D. melanogaster* (*Przewloka et al., 2007*), our own analyses are consistent with previous data (*Schleiffer et al., 2012*) that have revealed the presence of several kinetochore components in other insects, many of which appear to be highly divergent among insects. We therefore applied sensitive protein predictions using a combination of psi-blast, blastp, and tblastn in genomes, NCBI non-redundant protein databases, or within specialized proteomes (*Figure 1—source data.1*). Searches in the mRNA-Seq assemblies were performed using tblastn using human and any newly identified insect homologs. Hits were verified by identification of corresponding reciprocal best hits and HHpred prediction analyses.

## CenpS and CenpX

Using human and chicken CenpS and CenpX proteins as queries, we identified several insect homologs. We further verified those candidates using the structural based HHpred analyses aligning their conserved histone fold to known homologs. The annotation of the CenpX encoding gene in *B. mori* was not correct, but we could infer the correct transcript using *B. mori* EST data.

CenpT and CenpW homologs form a complex with CenpS and CenpX in kinetochores of vertebrates and fungi (*Perpelescu and Fukagawa, 2011*). CenpT and CenpW could not be found in any of the insect genomes or transcriptomes. We therefore conclude that CenpS and CenpX are conserved in insects due to their role in DNA repair (*Singh et al., 2010*), rather than as kinetochore components due to the absence of their binding partners CenpT and CenpW.

## CenpI

Vertebrate CenpI proteins reveal significant alignments to insect homologs that could be verified by reciprocal best-hit analyses. The CenpI encoding gene in *B. mori* genome was not correctly annotated, but we could correctly infer it using EST data. Three non-overlapping transcript fragments in the cricket assembly revealed significant CenpI alignments and both are included in *Figure 2—source data 1*. Homologs of the CenpI protein-binding partners, CenpH and CenpK, could not be identified in any insect genome.

## CenpL/M/N

CenpL and CenpM proteins appear to evolve rapidly, making cross-species prediction between insect and vertebrate homologs difficult. Still, insect homologs could be detected using annotated *Pediculus humanus corporis* CenpM and hymenopteran CenpL homologs and verified by HHpred and reciprocal best-hit analyses. CenpN homologs are unstructured proteins that rapidly evolve in animals and fungi. Nevertheless, insect CenpN homologs could be identified using psi-blast and verified by reciprocal best-hit analyses.

The N-terminus of CenpN interacts with CenH3 in humans (*Carroll et al., 2009*), and two arginine residues conserved across vertebrates have been shown to be important for CenpN–CenH3 interaction (*Carroll et al., 2009*). However, we could not identify any corresponding arginines in any of the insect CenpN homologs due to weak conservation between vertebrate and insect CenpN homologs. We therefore do not know if insect CenpN proteins retain the property of CenH3-binding, even in monocentric insects.

## CenpO/P/Q/R

The CenpO/P/Q/R complex is not essential in vertebrates (*Perpelescu and Fukagawa, 2011*); however recent findings in mice indicate that CenpU is essential in embryonic stem cells and in embryonic development (*Kagawa et al., 2014*). Homologs of the CenpO/P/Q/R complex could not be identified in any insect genome or transcriptome.

## Outer kinetochore proteins

We used *D. melanogaster*, vertebrate, and lepidopteran Mis12 and Ndc80 proteins to find their insect homologs. Both proteins were widely present in most but not all insects. Mis12 and Ndc80 are the central components of the outer kinetochore protein complexes. Using several rounds of psi-blast, we detected putative homologs of additional Mis12 complex components, Nnf1 and Dsn1, in insect genomes that also encode for Mis12 but not in the Mis12-deficient orders including the Coleoptera, the Hemiptera, and the Phthiraptera, supporting the loss of Mis12 in those orders. Still, experimental validation as performed in the *D. melanogaster* S2 cells (*Przewloka et al., 2007*) will be necessary to verify the presence of additional Mis12 complex components or confirm their absence. Ndc80 homologs could not be found in the stick insect and the cockroach transcriptome assemblies. We searched for Spc25 that is part of the Ndc80 complex and found homologs to be present in all insect genomes and transcriptomes, including the Ndc80-deficient stick insect and the cockroach assembly. Spc25 transcripts are highly abundant in all assemblies facilitating their detection. In contrast, Ndc80 abundance is more variable. Thus, it is possible that our inability to detect Ndc80 (*Figure 2*) was a result of lower expression, particularly since data only from one representative species was analyzed from either of these orders. It will be necessary to confirm this finding with analyses of additional representatives in the future.

## Phylogenetic analyses

Amino acid sequences of histone fold domains for CenH3 homologs or cupin domains of CenpC homologs were aligned using MUSCLE (*Edgar, 2004*). For *Figure 2D*, *Figure 1—figure supplement 3C*,

*Figure 1—figure supplement 4*, and *Figure 1—figure supplement 8B* the optimal amino acid substitution model was determined using prottest (*Abascal et al., 2005*). Maximum likelihood trees were generated using Phyml (*Guindon et al., 2010*). Bootstrap values supporting the topology with the highest maximum likelihood were obtained using Phylip (*Felsenstein, 1989*). For *Figure 1—figure supplement 6*, the maximum likelihood tree was built using the PhyML server (*Dereeper et al., 2008*) using default parameters and bootstraps set to 100. (*Suomalainen, 1966*; *Jarvis et al., 2005*; *Terry and Whiting, 2005*).

## Acknowledgements

We thank S Biggins, I Cheeseman, M Daugherty, L Kursel, A Marty, S Ramachandran, B Ross, F Steiner, P Talbert, J Young, and S Zanders for advice, helpful discussions, and comments on the manuscript. We are grateful to P Andolfatto for suggesting transcriptome assemblies as a search strategy, and to J Young and M Eickbush for help with transcriptome assembly analyses. We thank the Baylor I5K Project for providing access to four RNA-Seq data sets, and JC Walser for providing the *F. auricularis* assembly and early access to the transcriptome blast server. We would also like to acknowledge our colleagues S Nokkala, W Traut, F Marec, A Yoshido, and J Rufas for generously providing images of holocentric chromosome spreads for inclusion in our supplementary discussion. This study was supported by a postdoctoral Fellowship of the Jane Coffin Childs Memorial Fund for Medical Research (IAD), NIH grants R01-GM74108 (HSM). HSM and SH are Investigators of the Howard Hughes Medical Institute.

Sequence data are available in the NCBI Short Read Archive (SRA) (PRNJNA258192).

## Additional information

### Funding

| Funder | Grant reference number | Author |
| --- | --- | --- |
| Jane Coffin Childs Memorial Fund for Medical Research | Postdoctoral fellowship | Ines A Drinnenberg |
| National Institute of General Medical Sciences | R01 GM74108 | Harmit Singh Malik |
| Howard Hughes Medical Institute | Investigator | Steven Henikoff, Harmit Singh Malik |
| G Harold and Leila Y. Mathers Foundation | | Harmit Singh Malik |

The funders had no role in study design, data collection and interpretation, or the decision to submit the work for publication.

### Author contributions

IAD, Conception and design, Acquisition of data, Analysis and interpretation of data, Drafting or revising the article; DY, Acquisition of data, Contributed unpublished essential data or reagents; SH, HSM, Conception and design, Analysis and interpretation of data, Drafting or revising the article

### Author ORCIDs

Harmit Singh Malik, http://orcid.org/0000-0001-6005-0016

## Additional files

### Major datasets

The following dataset was generated:

| Author(s) | Year | Dataset title | Dataset ID and/or URL | Database, license, and accessibility information |
| --- | --- | --- | --- | --- |
| Drinnenberg IA, deYoung D, Henikoff S, Malik HS | 2014 | Insects Transcriptome or Gene expression | PRJNA258192; http://www.ncbi.nlm.nih.gov/sra/PRJNA258192 | Publicly available at NCBI Short Reads Archive. |

The following previously published datasets were used:

| Author(s) | Year | Dataset title | Dataset ID and/or URL | Database, license, and accessibility information |
|---|---|---|---|---|
| The modENCODE consortium | 2011 | Functional genomics project for the *Drosophila* modENCODE Project | http://www.ncbi.nlm.nih.gov/sra/SRX109311 | Publicly available at NCBI Short Reads Archive. |
| Roulin AC, Wu M, Pichon S, Arbore R, Kühn-Bühlmann S, Kölliker M, Walser JC | 2013 | De novo transcriptome hybrid assembly and validation in the European earwig (Dermaptera, Forficula auricularia) | http://www.ncbi.nlm.nih.gov/bioproject?LinkName=sra_bioproject&from_uid=557129 | Publicly available at NCBI Short Reads Archive, https://www.hgsc.bcm.edu/arthropods. |
| | N/A | Arthropod Sequencing at the BCM-HGSC | https://www.hgsc.bcm.edu/arthropods | Publicly available. |
| Zhen Y, Aardema ML, Medina EM, Schumer M, Andolfatto P | 2012 | Insect Genomics | http://genomics-pubs.princeton.edu/insect_genomics/data.shtml | Publicly available. |

**Reporting standards:** Standard used to collect data: Datasets reported as per guidelines of the NCBI Short Read Archive (SRA):

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
