## [Decision Letter]

Thank you for sending your work entitled “Recurrent loss of CenH3 is associated with independent transitions to holocentricity in insects” for consideration at *eLife.* Your article has been favorably evaluated by Diethard Tautz (Senior editor) and 2 reviewers, one of whom is a member of our Board of Reviewing Editors.

Your manuscript has been seen by two reviewers, both of whom are enthusiastic about your work. We would therefore like to publish your manuscript after you have addressed the following issue.

We feel that you are overstating the case with respect to the holocentricity of other less studied insect species. Although the text statements tend to be strong in papers about these species, the cytology often leads much to be desired. Obviously your data appear to correlate in that CenH3/CENP-A is lost is these species. For instance in the review by [55] and references therein, it is hard to find the original publications showing conclusive data regarding holocentricity. It would be useful if you could provide the key references where holocentricity has been clearly demonstrated for the species that they examine, or, failing this, to use more cautionary wording. We feel the manuscript would certainly be stronger if you had some cytology on other species, but this is not a requirement to publish.

A second issue is to explain the reoccurrence of holocentricity during insect evolution the authors “speculate that another, distinct event allowing CenH3 loss in holocentric insects must have occurred early in insect evolution”. They suggest that a lineage specific new protein may have arisen that allowed relaxation of the selective pressure for retention of CenH3/CENP-A. Have you used comparative analyses to try to identify 'new' proteins that are present in holocentric insect lineages but absent in more related or distant monocentric lineages? If not you could comment on this in your paper.

Minor comments:

1) The authors attribute centromeric histone variant sequences in two Lepidopteran assemblies to microsporidian contamination but refer to Figure 1—figure supplement 8, which only shows one locus. Was the same contaminant found in both assemblies?

2) The authors state that “Our findings suggest that holocentric insects have adopted a CenH3-independent inner kinetochore assembly pathway”; without any experimental analysis showing that the Cenp I,L,M proteins are localized to kinetochores in these species, this statement seems too strong. As with Cenp S/X, we do not know if these proteins only act at kinetochores or also in other contexts. We would suggest stating instead that “It will be important to test in the future if Cenp I,L,M localize to kinetochores in the absence of CenH3.”

3) The loss of Mis12 is quite intriguing given that it is emerging as the key linker between the chromatin components and the microtubule-binding activities. Did these same species lack the other Mis12 complex subunits? There was mention of 2 of the subunits in the Methods correlating with Mis12 – perhaps this could be better described in the primary text.

4) Similarly, loss of Ndc80 in cockroaches and stick insects despite presence of Spc25 was surprising. Is the Spc25 transcript count sufficiently high to be confident about this? These were not the focus on the authors' study but we are concerned that people will assume that the absences indicated were based on the same stringency analysis as employed for Cenp-A and Cenp-C

5) The authors cite the recent kinetoplastid work for an experimentally validated system lacking Cenp-A-based centromeres – they could also cite Monen et al (2005; PMID 16273096), who showed in *C. elegans* that outer kinetochore proteins could target to chromosomes in meiosis independently of Cenp-A (even though a Cenp-A chromatin foundation is employed in mitosis in the same species, which happens to be holocentric).

6) The authors state the CENP-O/P/Q/R/U are not essential in vertebrates. This is true in DT40 chicken cells but a recent knockout in mice of CENP-U by Kagawa et al (PMID 24481920) has challenged this view as CENP-U is essential in embryonic stem cells and in embryonic development (but not in embryo-derived fibroblasts; the reasons for this context-specific requirement are not yet known). This sentence could be reworded taking into account the newer findings.

---

## [Author Response]

*We feel that you are overstating the case with respect to the holocentricity of other less studied insect species. Although the text statements tend to be strong in papers about these species, the cytology often leads much to be desired. Obviously your data appear to correlate in that CenH3/CENP-A is lost is these species. For instance in the review by*
[55]
*and references therein, it is hard to find the original publications showing conclusive data regarding holocentricity. It would be useful if you could provide the key references where holocentricity has been clearly demonstrated for the species that they examine, or, failing this, to use more cautionary wording. We feel the manuscript would certainly be stronger if you had some cytology on other species, but this is not a requirement to publish*.

We agree with both suggestions. First, we now use more cautionary wording in the main text about the evidentiary nature of holocentricity in insects.

Second, prompted by the reviewers’ suggestions, we also have added a section in the Materials and methods that discusses some of the key pieces of evidence for holocentricity in insect orders, including original citations in all insect orders in which such a conclusion has been reached. In addition, we were fortunate that several colleagues (W. Traut, F. Marec, A. Yoshido) generously provided the original published and additional unpublished images of insect chromosome cytology from several species belonging to the Lepidoptera, and have given us permission to include these images in our supplemental material. Although we were unable to get equivalent cytological data on species belonging to the orders Phthiraptera and Dermaptera, we note that the conclusion of holocentricity in these original publications was based on a contrast with monocentric lineages that were also analyzed using similar methods and often by the same authors and research groups. Thus, although the photographic evidence in those early papers is hard to assess today, the author’s report of the observation could still be considered as being as authoritative as is currently possible. (We have incorporated this information in the main text within the Methods section.)

Finally, as the reviewers point out, our finding of CenH3 loss in each of the insect orders previously deemed holocentric also adds support to these original conclusions; CenH3 is present and essential in all monocentric insects tested.

*A second issue is to explain the reoccurrence of holocentricity during insect evolution the authors “speculate that another, distinct event allowing CenH3 loss in holocentric insects must have occurred early in insect evolution”. They suggest that a lineage specific new protein may have arisen that allowed relaxation of the selective pressure for retention of CenH3/CENP-A. Have you used comparative analyses to try to identify 'new' proteins that are present in holocentric insect lineages but absent in more related or distant monocentric lineages? If not you could comment on this in your paper*.

In our paper, we have focused on known kinetochore components of the CCAN complex, as present data would indicate these would be best placed to ‘replace’ CenH3 function in holocentric insects. As we discuss in the manuscript, the pattern of conservation of CCAN components does not reveal such a candidate. Although we think the suggestion of doing a detailed genomic analysis for genes conserved in holocentric versus monocentric insects is very interesting, two things affect this strategy. First, not all genomes of holocentric and monocentric insects are annotated well enough for us to be comprehensive with our searches. Second, although the expectation is that the protein that might have replaced CenH3 in holocentrics would be essential, there is no clear expectation that this protein would be non-essential and therefore lost in all monocentric insects (for instance, the CENP-B protein is non-essential in mammals, yet preserved in genomes). Instead, we are embarking on a proteomic analysis to try to identify the CenH3-replacing factor. We are at the initial stages of this analysis.

*Minor comments*:

*1) The authors attribute centromeric histone variant sequences in two Lepidopteran assemblies to microsporidian contamination but refer to*
Figure 1—figure supplement 8*, which only shows one locus*. *Was the same contaminant found in both assemblies?*

We thank the reviewers for bringing this to our attention. We now clarify this in our revision. We found *CenH3* contaminants from different microsporidian in two lepidopteran assemblies. First, we found a *CenH3* transcript in the *Cycnia tenera* assembly that we can confidently attribute to being derived from the microsporidian *Nosema bombycis* (Figure 1—figure supplement 8). Second, we found a *CenH3* transcript in the *Papilio glaucus* assembly, which we refer to as Locus 1862 (transcript 3) (Figure 1—figure supplement 8). Although we don’t know the source of this transcript, we are confident it is of microsporidian origin based on the phylogenetic analysis. We now clarify both these points by more extensive annotation of Figure 8B.

*2) The authors state that “Our findings suggest that holocentric insects have adopted a CenH3-independent inner kinetochore assembly pathway”; without any experimental analysis showing that the Cenp I,L,M proteins are localized to kinetochores in these species, this statement seems too strong. As with Cenp S/X, we do not know if these proteins only act at kinetochores or also in other contexts. We would suggest stating instead that “It will be important to test in the future if Cenp I,L,M localize to kinetochores in the absence of CenH3*.*”*

We edited our manuscript as suggested.

*3) The loss of Mis12 is quite intriguing given that it is emerging as the key linker between the chromatin components and the microtubule-binding activities. Did these same species lack the other Mis12 complex subunits? There was mention of 2 of the subunits in the Methods correlating with Mis12 – perhaps this could be better described in the primary text*.

Unfortunately, of all the components of the Mis12 complex, Mis12 is the easiest to bioinformatically detect, whereas the other subunits appear to be very difficult to predict due to low protein sequence similarity and lack of characteristic protein domains. In fact, *D. melanogaster* homologs could only be identified in mass spectrometry studies (PMID: 17534428). Although we could find putative homologs of these other subunits in Lepidoptera and Hymenoptera but not in any of the Mis12-deficient species, we are wary of making a stronger conclusion about this based on the present analyses. We believe that experimental validation will be necessary to characterize the composition of the Mis12 complex and conclude the presence or absence of complex members other than Mis12 as it was performed in *D. melanogaster*.

*4) Similarly, loss of Ndc80 in cockroaches and stick insects despite presence of Spc25 was surprising. Is the Spc25 transcript count sufficiently high to be confident about this? These were not the focus on the authors' study but we are concerned that people will assume that the absences indicated were based on the same stringency analysis as employed for Cenp-A and Cenp-C*.

We indicated Spc25 transcript abundance in Figure 1 and Figure 1—figure supplement 5. In most assemblies the transcript is within the upper 90^th^ percentile of abundance. Furthermore, Spc25 homologs appear to be very similar to one another (in contrast to its interaction partner Spc24) facilitating predictions even without full-length protein sequence alignments.

Ndc80 homologs can be easily predicted based on protein sequence similarity. At least based on this, we are confident about our inability to detect Ndc80. However, our conclusion that Ndc80 is indeed absent in Phasmatodea and Blattodea will be greatly increased with at least one additional species from each order. We therefore use more cautious wording about the absence of Ndc80 in stick insects and cockroaches.

*5) The authors cite the recent kinetoplastid work for an experimentally validated system lacking Cenp-A-based centromeres – they could also cite Monen et al (2005; PMID 16273096), who showed in C. elegans that outer kinetochore proteins could target to chromosomes in meiosis independently of Cenp-A (even though a Cenp-A chromatin foundation is employed in mitosis in the same species, which happens to be holocentric)*.

We have now included this reference in our revised version and thank the reviewers for this suggestion.

*6) The authors state the CENP-O/P/Q/R/U are not essential in vertebrates. This is true in DT40 chicken cells but a recent knockout in mice of CENP-U by Kagawa et al (PMID 24481920) has challenged this view as CENP-U is essential in embryonic stem cells and in embryonic development (but not in embryo-derived fibroblasts; the reasons for this context-specific requirement are not yet known). This sentence could be reworded taking into account the newer findings*.

We included the reviewers’ comment in our revised version and thank them for this suggestion.